



# The effect of site-specific wind conditions and individual pitch control on wear of blade bearings

Arne Bartschat, Karsten Behnke and Matthias Stammler

Fraunhofer Institute for Wind Energy Systems IWES, Am Schleusengraben 22, 21029, Hamburg, Germany

*Correspondence to*: Arne Bartschat (arne.bartschat@iwes.fraunhofer.de)

**Abstract.** The characteristics of a pitch controller determine how the wind turbine reacts to different wind conditions. Control strategies like individual pitch control are known for their ability to reduce the amplitudes of load cycles of the structures of the wind turbine while influencing the operation conditions of the blade bearings in a challenging way. However, the control strategy is not the only influencing factor with respect to failure modes of blade bearings like wear and raceway fatigue. The

site-specific and stochastic wind conditions can cause wear-critical operating conditions, which are usually not reflected in the rather short time frames of aeroelastic simulations.

This work analyses wind and operating conditions regarding their influence on wear in blade bearings. It is based on measured wind conditions and the modelled behavior of the IWT-7.5-164 reference wind turbine with respect to its pitch activity. The simulation data is used to determine the longest period of uninterrupted wear-critical operation and create a test program based

on it for scaled and real-size blade bearings. Experimental results based on this test program show that wear-critical operation conditions can occur during normal operating of a wind turbine and cause wear damage to the bearing raceways.

## 1 Introduction

Blade bearings enable the rotation of the rotor blades along their longitudinal axis to change the angle of attack of the inflow. This rotational movement is called pitching and is a fundamental feature of modern, pitch-controlled wind turbines. The

collective pitch controller (CPC) of wind turbines limits the rotational speed of the rotor above rated wind speed (Burton, 2011). This ensures a reduction of loads and allows the wind turbine to operate at higher wind speeds. In addition, load mitigating control strategies like individual pitch control (IPC), as part of the pitch controller, can lower the fatigue relevant structural loads under all operation conditions while reducing the annual energy production (AEP) only to minor extent (e.g. (Barlas and van Kuik, 2010), (Requate et al., 2020)). However, implementing such control strategies will amplify both the

number of cycles and the amplitudes of the oscillating movements of pitch bearings, which could cause the occurrence of wear as well as accelerating rolling contact fatigue (Stammler et al., 2020). A pitch controller development aims at a compromise between the risk of pitch bearing failures, structural loads and AEP.

While structural loads usually can be assessed with methods and simulations described in the IEC 61400-1 (Wind energy generation systems, 2019), there is no straightforward or generalized method for evaluating the wear risk known to the authors.



Wear only occurs in a rolling contact bearing under starved contact lubrication conditions.

The wind turbine design guideline 03 (DG03, (Harris et al., 2009)) suggests that oscillation angles of $\theta < \theta_{crit}/2$ promote the risk of fretting corrosion and therefore the occurrence of wear in form of false brinelling. $\theta_{crit}$ is the oscillation angle, at which the contact track gets in touch with two rolling elements. For smaller oscillation angles like $\theta_{dith}$, which equals a cyclic movement of the Hertzian contact smaller than its width, a standstill mark is likely to occur (Grebe et al., 2018). To avoid such

wear promoting operating conditions it is suggested to incorporate larger movements with $\theta > \theta_{crit}$ to redistribute the grease on the raceways and therefore relubricate the contact areas to prevent starvation. Tests on small scale bearings performed by showed that interrupting oscillating and wear critical movements with larger movements effectively prevents wear. This was even tested and verified for real-scale blade bearings with about 5 m diameter (Stammler (2020), Behnke and Schleich (2022)). The change of the pitch angle of wind turbine will lead to changes of the operation point and loads acting on the wind turbine.

Therefore, the incorporation of larger oscillation angles, often referred to as lubrication runs, in the operation of the wind turbine is related to compromises between loads, energy yield and damage modes of the blade bearing and pitch systems (Schwack et al., 2018). As this compromise is hard to find because some damage modes like wear of blade bearings are still not completely understood, pitch controllers do not yet support on-demand lubrication-run strategies to the knowledge of the authors. However, pitch controllers of turbine manufacturers usually execute lubrication runs on a regular basis like once every

24 h to follow suggestions of the bearing manufacturers and to lower the risk of wear on the raceways. The work related to this paper focuses on rather short time frames of less than 24 h to analyze the risk of wear damage of the pitch bearings under normal operating conditions. Stammler (2020) developed methods for evaluating load time series data regarding wear critical oscillations. Load time series data usually consist of continuous intervals of about 600 s which is too short to accumulate wear-critical operation conditions which provoke wear. Considering that the rotational speed of the turbine determines the number

of oscillations in a specific time frame, it is conceivable that wear-critical operation conditions will form over hours and days which will be called medium-term behavior in the following. To mitigate this disadvantage of load time series data, Stammler (2020) uses a generic stitching approach for wear influencing operating conditions consisting of critical and beneficial sequences of pitch activity in order to develop a test program for wear of blade bearings. The aim of this test program is to replicate all wear critical operation conditions occurring over the turbine lifetime. The order of occurrence of the critical and

beneficial sequences of oscillations is based on seasonal variation in the mean wind speed (long-term). Due to the nature of the compilation of beneficial and critical sequences the test program probably does not directly reflect medium -term real-world wind turbine operation. The study in this paper is not aiming at developing a new endurance test program covering the lifetime of the wind turbine. Instead, it intends to show that even without applying advanced stitching or rearranging algorithms to sequences of loads and pitch movement patterns, it is possible to accumulate wear critical operating conditions during

normal operation in a rather short amount of time. Starved contact lubrication conditions especially can occur under oscillating operating conditions like in blade bearings (Wandel et al. (2022)). While the Hertzian contact travels over the raceway in an oscillating movement pattern, the lubricant which separates the contact partners is pushed out of the contact area and out of



the contact track. This phenomenon has been under intensive investigation in the research projects like HBDV, HAPT and iBAC.

Behnke and Schleich (2022) list different publications with respect to wear dedicated test programs and the amount of cycles in several tests. Usually, numbers of about 40,000 oscillations with generic amplitudes and frequencies are tested to investigate wear as a failure mode. However, it is rather unlikely for a wind turbine to accumulate 40,000 wear critical cycles without changing the mean angle, frequency, or amplitude and hence to not include wear preventing cycles. Therefore, Schwack et al. (2022) also looked into wind conditions and load time series data and found out that worst case operation conditions are able

to accumulate well above 13,000 wear critical oscillation cycles. They then conducted tests with 13,000 cycles, but were limited to consider variations in amplitudes, frequency, or mean values in the tests. Stammler et al. (2019) performed several tests on angular contact ball bearings of type 7220 to show the influence of random movement pattern variations and the effectiveness of lubrication-runs in terms of protecting the bearings from wear. However, being the first investigation showing oscillation angle variations and the effectiveness of lubrication-runs, the work does not answer the question to what extent and

in what order protection runs and wear critical cycles can be found in real world wind turbine operation.

The present study uses the same wind data used in Schwack et al. (2022) in combination with an aeroelastic model of the IWT-7.5-164 reference wind turbine to obtain a better understanding on worst-case operating conditions with respect to wear of blade bearings. The novelty of the approach presented in this paper is the consideration of the order of occurrence of the pitch movements and the representation of a realistic behavior of a wind turbine.


The remainder of this paper is structured as follows: Section 2 explains the wind data analysis with the focus on finding wear critical operation conditions. Section 3 describes the simulations with an aeroelastic wind turbine model and site-specific wind conditions. The compilation of a test program based on the wind data analysis and the simulation results as well as the test setup are described in Section 4. The results of experimental assessment of this test program with scaled and real-size blade

bearings are presented in Section 5 and discussed in Section 6. Section 7 concludes the findings of this paper.

## 2 Wind data analysis

This chapter gives an overview of the measured wind conditions used in this paper. In addition, the data set is analyzed with the aim of finding wear critical operating conditions.

### 2.1 Wind data characterization

The dataset consists of one year of 10-minute values of LIDAR (light detection and ranging) measurements at a nearshore site. The wind conditions and site properties match the design criteria of the IWT7.5-164 reference turbine.

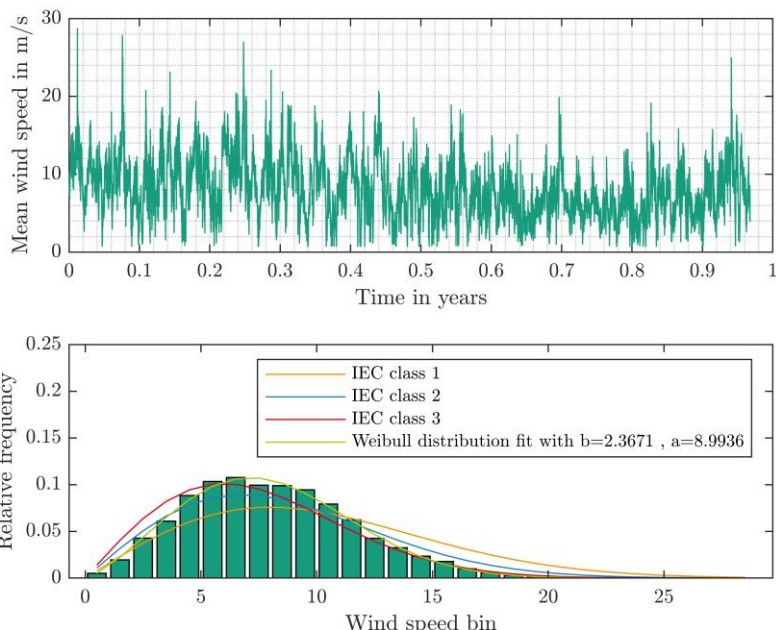

**Figure 1: Measured mean wind speeds (top) and relative frequency of measurements compared to wind classes defined in the IEC-61400-1 (bottom)**

The LIDAR measurement data contains information for heights of 119, 139, and 159 m and consists of wind speed mean values, maximum values, and standard deviations. It also contains values of the wind direction and the turbulence intensity. Figure 1 shows the measured mean wind speeds (top). A bin sorting (Figure 1, bottom) reveals the wind speeds do not correlate well with any of the IEC wind classes (Wind energy generation systems, 2019). As the best fit would be wind class 3, the site is generally characterized by rather low wind speeds.

Figure 2 shows the mean turbulence intensity per bin and adds information for mean plus (red asterisk) and minus (green asterisk) the corresponding standard deviation to indicate the width of the distribution of measurements. The turbulence intensity (TI) of the measurements is rather low compared to the NTM suggested by the IEC-61400-1 and the different turbulence classes (Wind energy generation systems, 2019). The turbulence intensity at this site is well below class C for almost all wind conditions.



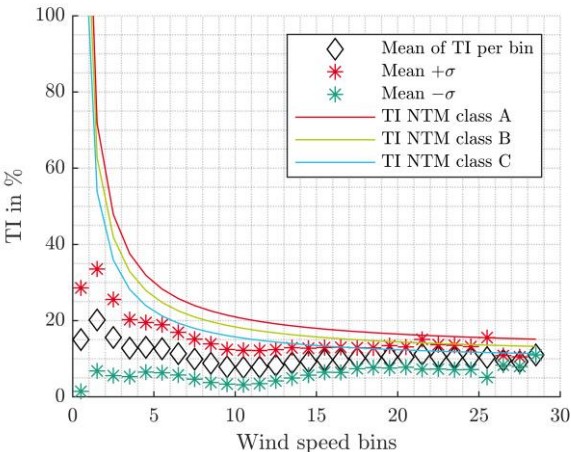

**Figure 2: Turbulence intensity (TI) per bin derived from wind measurement data compared with the characteristics of normal**
**turbulence model (NTM)**

As the data contains measurements for three different heights, it is possible to estimate the vertical wind shear exponent α of the normal wind profile model suggested by the IEC 61400-1 (Wind energy generation systems, 2019). Therefore, the wind speed measurements for each time step and for the three different heights are used to fit the power law function in Eq. 1.

$$V(z) = V_{hub} \cdot \left(\frac{z}{z_{hub}}\right)^{\alpha} \qquad (1)$$

The exponent α which leads to the lowest error between power law function and measured data is used as an estimation for the shear. Figure 3 shows the obtained results for α per bin and reveals that the site has a stronger vertical wind shear than the IEC standard value of 0.2 for almost all wind speeds.

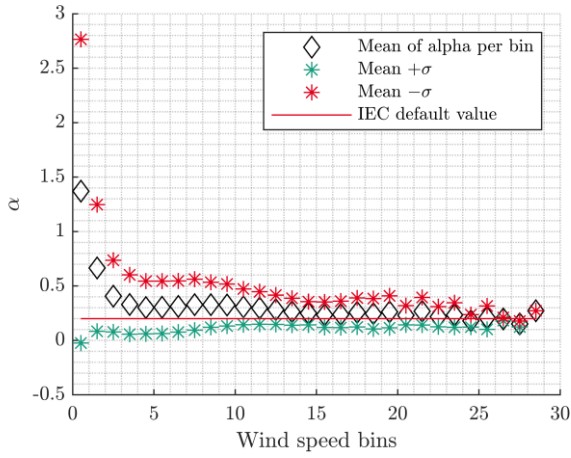

**Figure 3: Wind shear exponent α of the normal wind profile model derived from wind measurement data**





## 2.2 Wind data and turbine operation conditions

The characteristics of the measured wind speeds explored in the foregoing chapter have the following influences on the wind turbine operation: The lower turbulence intensity will lead to fewer load cycles, less rotational speed variation and less dynamic operating conditions than indicated by the IEC turbulence classes. However, as the wind shear is higher than stated by the

standards, the load cycles especially for the out-of-plane bending moments of the rotor blades will have larger amplitudes. As the wind turbine operates with IPC, the controller will lead to larger oscillation angles and more activity of the pitch system and the blade bearing. While the shear induces more pitch activity, the low TI leads to less variation in the oscillations of the bearings. It is therefore reasonable to assume a more wear-critical operation under the measured wind speeds in comparison to IEC classes. Aero-elastic simulations done for IEC classes will not reflect the site-specific operational conditions.

Like the low TI which reduces short-term variations of the wind speed, steady wind speeds over several 10-minute intervals can result in aggregation of wear-critical conditions. Therefore, the wind measurement data is checked for longer periods of time without major changes in the mean wind speeds. Especially for wind speeds below rated and IPC activity, many oscillation cycles without much deviation of the oscillation amplitudes and without changes of the mean oscillation angle will occur. The used algorithm counts consecutive wind speeds within a certain wind speed bin and a predefined window. For each data point

(i.e. 10-minute interval), the algorithm will count the number of consecutive measurements within the same or the directly neighbored wind speed bins.

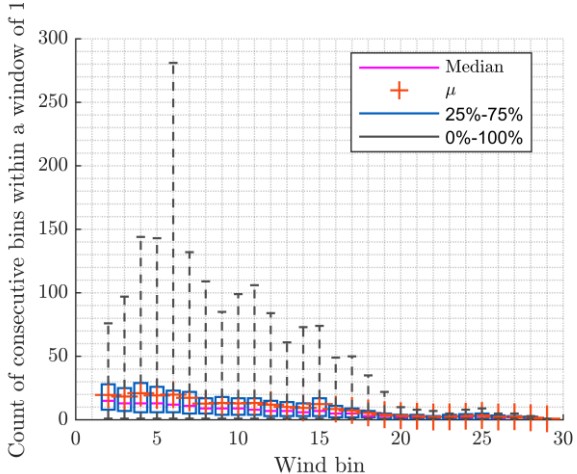

**Figure 4: Results of counting of consecutive occurrence of wind speed mean values**

Figure 4 shows the results of the algorithm with respect to window of plus and minus one bin (1 m/s). The figure indicates the

complete distribution of the results per wind speed bin using a boxplot. The maximum values, or upper whiskers, are of interest for the further analysis as they show the longest periods.

Not all periods will lead to wear-critical operation. At very low wind speeds, the loads are too low to induce wear. At wind speeds above rated wind speed, CPC operation will cause variations in mean values of oscillations and hence reduce the wear risk. Close to and below rated wind speed, the turbine operates with high loads and almost no CPC activity. Especially for the




controller and turbine configuration used for this work, the partial load regime between 8 and 10 m/s wind speed is
characterized by IPC activity without any CPC. As the algorithm with its counting window set to plus and minus one bin (1
m/s) incorporates the neighboring wind speed bins, the bin of 10 m/s includes some data of 11 m/s which already can be
characterized by CPC activity. The bin at 9 m/s, however, is well within the before mentioned partial load regime. It has a
maximum number of consecutive 10-min intervals of 82. This means that the mean wind speed stayed around 9 m/s for about

13.7 h of operation. Figure 5 shows the interval of mean measured wind speeds for this specific period which will be called
the critical sequence in the following. The vertical green lines indicate the start and end of the critical sequence to show how
it is embedded into an even longer time frame. (Schwack et al., 2022) followed a slightly different approach as they
incorporated periods of slower wind speeds into the critical sequence as well. Figure 6 and Figure 7 show the turbulence
intensity and the shear exponent α for the same critical interval.


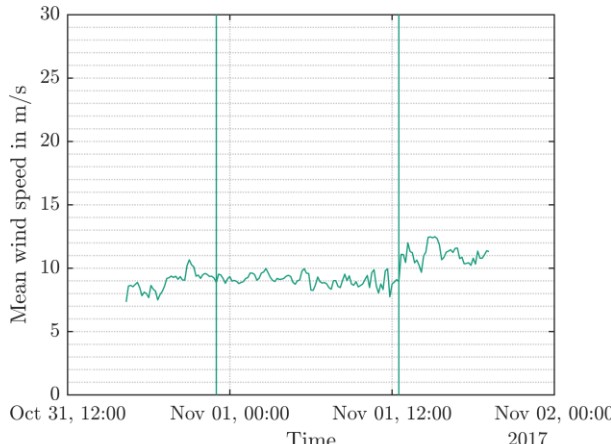

**Figure 5: Interval of longest consecutive measured mean wind speeds around the wind speed bin of 9 m/s**

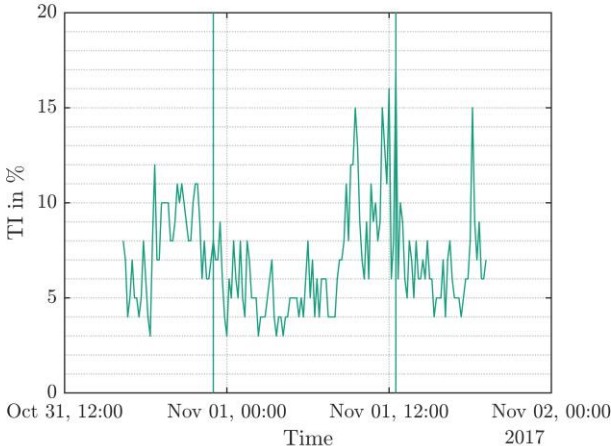

**Figure 6: Turbulence intensity within the interval of consecutive measured mean wind speeds around wind speed bin of 9 m/s**

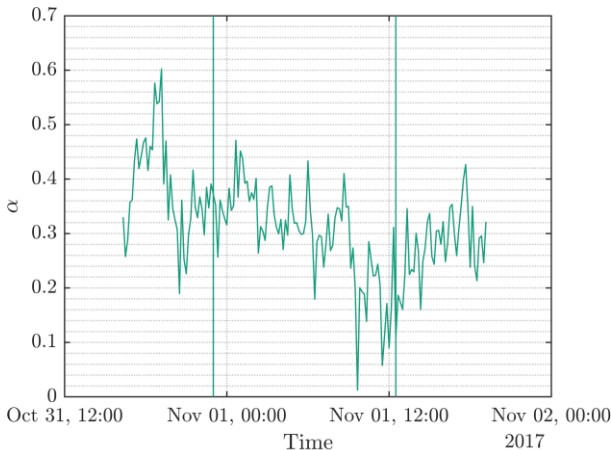

**Figure 7: Normal wind profile model shear exponent alpha within the interval of consecutive measured mean wind speeds around wind speed bin 9 m/s**

The mean wind speeds, turbulence intensity and shear exponent α are used as an input for aeroelastic simulations and the
compilation of the test program which will be explained in the next chapters.

**3 Aero-elastic simulation with measured wind conditions**

To check if the operating conditions described in the previous chapter can be considered critical in terms of wear, simulations
with the IWT-7.5-164 reference wind turbine are carried out. The turbine has a rotor diameter of 164 m, a hub height of 119.3
m and a rated power of 7.5 MW. It is designed for nearshore sites and wind class IEC A1. Further information can be found
in (Stammler et al., 2020). The mean wind speeds, turbulence intensities, and power law exponents of each 10-minute interval
in the critical sequence are used as an input for generating individual wind fields with a duration of 600 s. These wind fields





use the normal turbulence model and are generated with TurbSim (Jonkman and Kelley, 2021). To reflect the stochastic nature of the wind, six different seeds are created for each 10-minute interval. This results in a total of 492 wind fields with 82 mean

wind speeds and six seeds. The wind turbine model is simulated using those 492 wind fields to characterize the specific behavior of the turbine model under these operating conditions. The simulations are carried out using The Modelica Library for Wind Turbines (MoWiT) (e.g. Thomas (2022), Popko et al. (2021)). The software is a tool for fully coupled aero-hydro-servo-elastic simulations of wind turbines. It has been verified against other commercially available and commonly used simulation software at Fraunhofer IWES. The simulations are carried out using aa research controller developed by Fraunhofer

IWES. The controller is tuned to mimic the behavior of an industrial pitch controller described in Stammler (2020). A complete set of simulated design load cases with the industrial controller is available at (Popko (2019)).

In general, the controller incorporates CPC and IPC. The IPC activity is slightly reduced under partial load conditions to lower the oscillating pitch activity. This behavior can be described as a compromise between pitch activity and load mitigation, as it reduces pitch activity under wind speed conditions where no high fatigue relevant loads will occur. Besides this, the controller

has a lower boundary of the pitch angle at approximately 0°. This boundary is motivated by system restrictions and further reduces the IPC activity. Figure 8 shows how the controller changes the pitch angle as a reaction to a stepwise increase of wind speeds. This indicates the general behavior and characteristics of the controller.

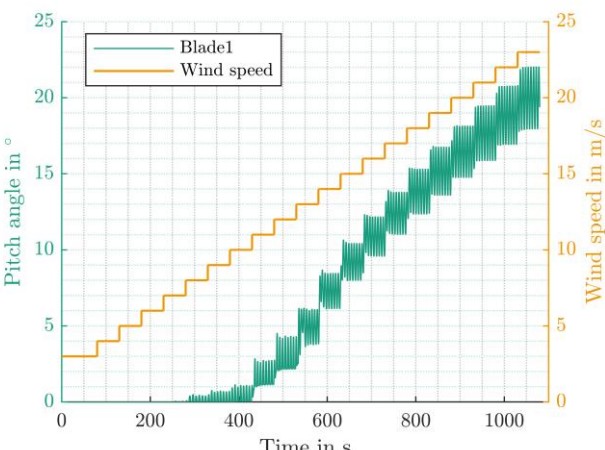

**Figure 8: Stepwise increase of simulated wind speed and pitch angle for the IWT-7.5-164 turbine and controller**

Besides a wind turbine model and an IPC controller, some bearing specific information is needed for setting up a test program. A blade bearing was designed for the IWT-7.5-164 reference turbine (Stammler, 2020). The bearing is a 4-point contact double row ball bearing, which is a very common topology for blade bearings. Table 1 lists the major properties of this bearing and of the other bearings used for experiments described in this paper. The bearing properties are used for scaling of time series data to different bearing dimensions.





**4 Simulation results and test program**

This chapter explains how the test program is derived from the simulation results. It also describes the scaling of the test program to the different bearing sizes.

**4.1 General description of the test program and simulation results**

To evaluate if the critical sequence described in Chapter 2 is in fact critical in terms of wear, a test program has been developed
using the simulation results of Chapter 3. These simulation results are time series for pitch angles and loads of a blade bearing. The test program should reflect the 13.7 h of turbine operation based on the simulated time series data without any further processing inside the individual time series. For each of the 82 steps within the critical sequence, one of the six simulated seeds per step is selected for the test program. The selection is determined by the pitch movement patterns which are evaluated with a wear rating factor which is described in the following.
Stammler et al. (2018) suggested range-pair counting for analyzing the pitch movement cycles which is also used for this investigation. The 492 simulated 10-minute intervals are characterized by the amount of full pitch movement cycles without a considerable change in the mean pitch angle. Figure 9 shows the simulated pitch angle $\theta$ for one blade within two individual 10-minute intervals of the critical sequence of operating conditions. The red line at the beginning and the blue line at the end are markers for the pitch angle interval considered for the counting of full pitch movement cycles. The red and blue diamond
markers are referring to upper and lower turning points of the pitch angle. A full cycle is composed of a movement between three turning points of the pitch angle $\theta$ – e.g. from lower to upper and back to lower turning point. The distance between the individual turning points describes the double amplitude or the peak-to-peak value of an oscillating movement. Hence, a full pitch cycle can also be described by two consecutive double amplitudes. The orange line in the graphs shown in Figure 9 indicates the mean pitch angle of the detected double amplitudes.






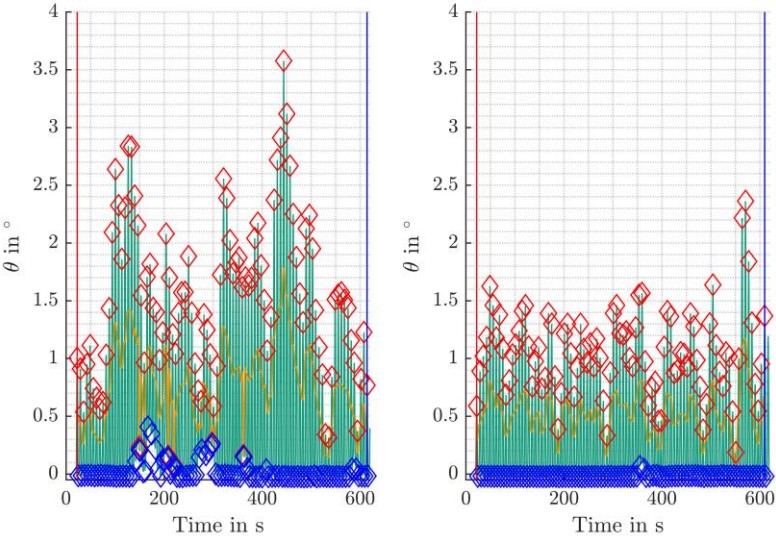

**Figure 9: Simulated pitch angle ($\theta$) time series of two individual 10-min intervals within the critical sequence of operating conditions.**

Although the two time series stem from comparable wind speed bins, the pitch angle time series are quite different. The left graph has considerably more fluctuation in the pitch angle signal, higher maximum values, and even some CPC activity at
about 180 s and 280 s. However, the number of full cycles is almost the same as in the right graph, due to the same rotational speed of the turbine throughout the simulation. The overall appearance of the pitch angle time series on the right is much more homogeneous with less fluctuation and with lower maximum values.

Each simulated 10-min interval is characterized by the following values:

-    Number of full pitch cycles
-    Maximum pitch distance
-    Standard deviation of double amplitudes

The basic principle of those values with respect to wear can be characterized as follows. High numbers of full pitch cycles can be considered worse than low numbers. However, as the wind speed conditions within the critical sequence are almost constant, the rotational speed of the wind turbine does not change to a large extent. As IPC mitigates cyclic bending moments acting on
the blade roots and these cyclic loads are depending on the rotational speed of the rotor, the amount of pitch cycles per 10-minute interval is almost equal for the 82 steps. The number varies around 90 cycles per 10-minute interval. Hence, the number of full cycles is not considered as a selection criterion. The maximum pitch distance is calculated as the difference between the lowest lower and highest upper turning point. Larger pitch distances have better chance to enhance the lubrication conditions in the contact due to grease redistribution (Wandel et al. (2022)). Hence, lower pitch distances can be considered
worse. The last selection criterion is the standard deviation of double amplitudes. Smaller values of the standard deviation are characterizing less variation in the pitch angle distribution and therefore a more monotonic sequence of pitch cycles which is





considered more critical in terms of wear. The maximum pitch distance ($\Delta\theta_{max}$) and the standard deviation of the double amplitudes ($\sigma_{double\ amplitude}$) are combined to form a single wear rating factor ($WR_{seed}$) for each seed (see Eq. (1)):

$$WR_{seed} = \frac{1}{\sigma_{double\ amplitude}} \cdot \frac{1}{\Delta\theta_{max}} \qquad (2)$$

The seed with the maximum wear rating factor is chosen for each of the individual 82 steps within the critical sequence. The concatenation of those worst-case operating conditions following the given order of the critical sequence forms the basis of the test program. The individual steps are connected with ramped transitions of about 5 s duration. These ramps are created in a way that high accelerations and jerks are avoided to ensure a smooth transition in the test program. Figure 10 shows an overview of the individual steps from the wind time series analyses to the final test program.

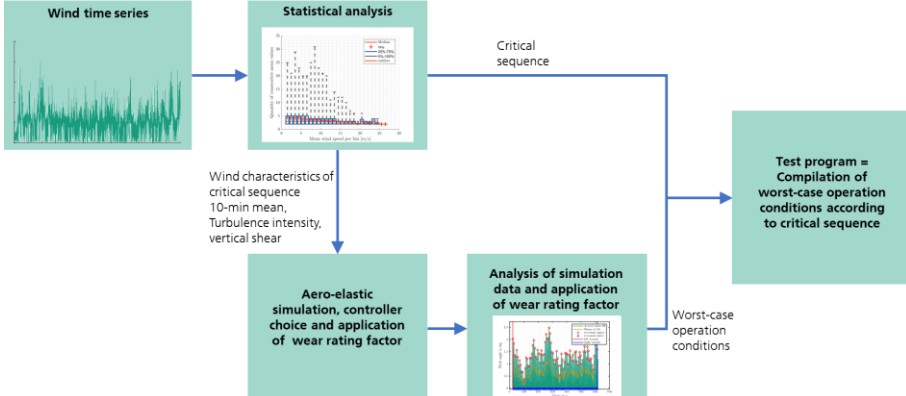


**Figure 10: Test program generation using site-specific wind speed conditions and wind turbine simulation data**

The final test program has a total length of 13.7 h and incorporates a total of 7,500 pitch cycles without considerable change of the mean pitch angle.

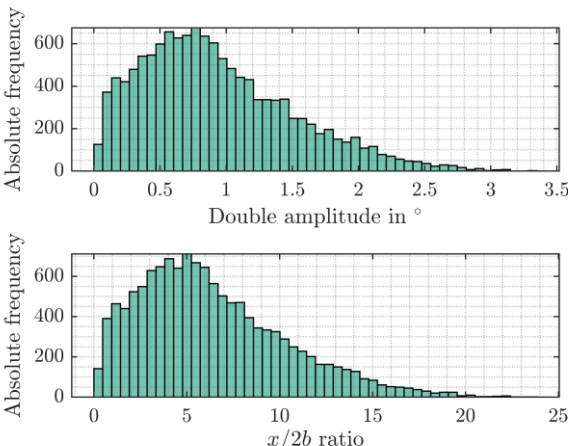

**Figure 11: Absolute frequency of double amplitudes (top) and corresponding x/2b ratios (bottom) in the critical sequence of operation conditions**



Figure 11 shows the absolute frequency of double amplitudes and corresponding x/2b ratios in the critical sequence of operation conditions and the test program. x is the travelled distance of the rolling element on the raceway between the upper and lower turning point of the oscillation and 2b is the width of the contact ellipse parallel to the direction of rolling. The x/2b ratio is often used to describe the operating conditions in wear related tests. The most frequent x/2b ratio within the test program is about 5, while the greatest value can be as high as 25. The corresponding ranges of the double amplitudes are about 0.7° for the most frequent angles and 3° for the maximum angle. These values are in the range of test conditions mentioned in various publications and are expected to be sufficient to produce wear according to Wandel et al. (2023).

**4.2 Bearings used for the experiments**

To evaluate the test program with respect to the ability of creating wear on the raceways, tests on different test rigs available at Fraunhofer IWES are carried out. Table 1 gives the bearing parameters for all simulated and tested bearings. The smallest bearings in this investigation are angular contact ball bearings of type 7220 tested on the BEAT0.2 (Bearing Endurance and Acceptance Test Rig). The midsize bearings are downscaled blade bearings with an outer diameter of 750 mm at the BEAT1.1. These bearings are smaller but share the same topology as the larger four-point contact ball bearings (4pBB) used as blade bearings. In addition, the manufacturing processes are the same as for real scale blade bearings. Hence, the surface finish, hardening processes and the overall stiffness of the bearings are comparable to real world applications. The largest bearings are real scale blade bearings. These blade bearings are 4pBB with an outer diameter of 2.6 m and 4.69 m. The smaller ones stem from a commercial turbine with a rated power of 3 MW and are tested on the BEAT2.2 (see also Behnke and Schleich (2022)). The larger one is designed as a reference bearing for the only virtually existing IWT-7.5-164 reference turbine. However, the same design has been used for tests at the BEAT6.1 test rig within the HAPT project. Further information for this specific bearing can be found in (Stammler, 2020).

All bearings are lubricated with commercial greases, which are also used or especially recommended for blade bearings. All bearings used for experiments described in this paper are lubricated with a grease which has a lithium-based thickener and a base oil viscosity of 50 cSt at 40°C (referred to as grease no.1). In addition, a second grease is used with the downscaled blade bearings in an additional test to point out its impact on wear formation. This grease also has a lithium-based thickener but uses a base oil with a higher viscosity of about 460 cSt at 40°C (referred to as grease no.2).





**Table 1: Properties of tested bearings**

| Property | Sign | IWT-7.5-164 blade bearing (4pBB) | Angular contact ball bearing of type 7220 | Downscaled blade bearing (4pBB) | Real size blade bearing for 3MW platform (4pBB) |
|---|---|---|---|---|---|
| Abbreviation | | 12480 | 7220 | 13229 | 51818 |
| Pitch diameter | $dm$ | 4690 mm | 139.809 mm | 673 mm | 2310 mm |
| Ball diameter | $D$ | 80 mm | 25.4 mm | 25.4 mm | 65 mm |
| Initial contact angle | $\alpha$ | 45° | 40° | 45° | 40° |
| Number of balls per row | $z$ | 147 | 15 | 69 | 98 |
| Number of rows | m | 2 | 1 | 2 | 2 |
| Groove conformity | $f_i$ | 0.5319 | 0.522 | 0.532 | 0.53 |
| Gamma (Harris and Kotzalas, 2006) | $\gamma = \cos(\alpha) \cdot D/dm$ | 0.0121 | 0.1392 | 0.0267 | 0.0216 |
| Inner raceway diameter | $di = dm - \cos(\alpha) \cdot D$ | 4633.4 mm | 120.35 mm | 655.04 mm | 2260.2 mm |

## 280  4.3 Test setups for experimental investigations

This chapter gives an overview of the test rig infrastructure used to test the bearings described in the previous chapter. All test rigs are part of the bearing test infrastructure at the Fraunhofer IWES.

### 4.3.1 Test setup for type 7220 bearings

The bearings of type 7220 are angular contact ball bearings, and they are used for several fundamental investigations regarding
285  raceway wear under oscillating operating conditions (e.g. Stammler (2020)). Figure 12 shows the BEAT0.2 with its main components. The test rig is equipped with a servo motor for accurate position control, a torque meter, a hydraulic load application system, and a shaft holding two bearings. The bearings are loaded with a static axial load and operated with a horizontal rotation axis.



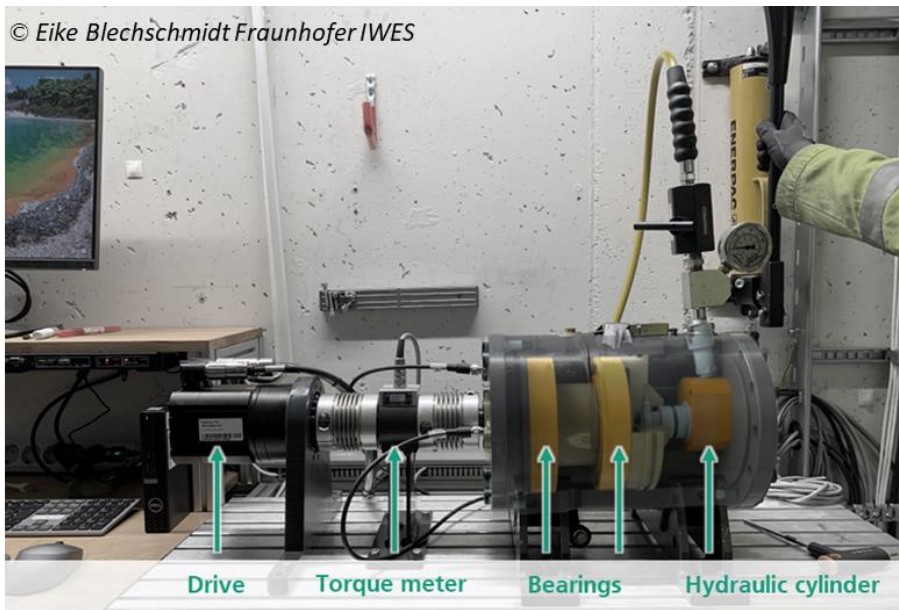

**Figure 12: Test rig BEAT0.2 for type 7220 bearings**

As the test rig can only apply static loads, the contact pressure and the contact angle are static as well. This is considered in the scaling of the test program (see Chapter 4.4). The axial load for the test is set to 90 kN, which leads to a Hertzian contact pressure of about 2.5 GPa and a contact width $2b_{7220}$ of about 0.9 mm. This contact pressure is also present in the simulated operating conditions of the IWT-7.5-164. Each bearing is cleaned with a water-based cleaning agent, dried with pressurized air, and lubricated with about 160 ml of grease no. 1. The bearings are run in for about 100 rotations back and forth under load to distribute the grease within the bearings before the test.

### 4.3.2 Test setup for type 13229 double row ball bearings

The BEAT1.1 test rig is shown in Figure 13. The test rig uses a hexapod construction and can apply loads in six degrees of freedom. Hence, it is particularly suited to load the tested bearings in the same way as blade bearings are loaded in a wind turbine. The test rig tests two bearings at the same time in a back-to-back configuration. Both inner rings of the bearings are connected to a so-called force transmitting element which also is connected to a gear which is driven by an electrical pitch system. The inner rings and force transmitting element share the same vertical rotational axis in this test rig. Additional information regarding this test rig can found in (Menck et al., 2022) and (Graßmann et al., 2023).





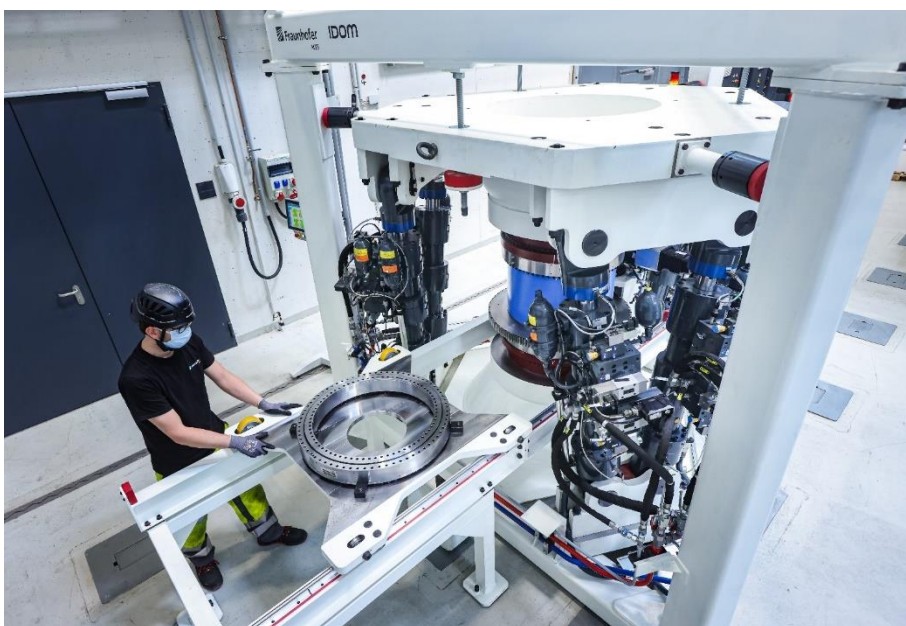

**Figure 13: BEAT1.1 test rig and downscaled blade bearing 13229 (©Fraunhofer IWES/ Ulrich Perrey)**

The bearings are equipped with 600 ml of either grease no.1 or grease no.2 and subjected to a comprehensive run-in procedure after installation in the test rig. Before the wear dedicated test program, a sequence of different friction torque related test programs has been executed on this test rig. These test programs have been performing a total of 454 movements with ±60° at different rotating speeds and load conditions for evaluating the friction torque behavior of each set of bearings. In addition, these test programs have ensured a proper grease distribution and run-in procedure for the test bearings before the start of the wear-related tests.

### 4.3.3 Test setup for blade bearings type 51818

Figure 14 shows a photo composition of the BEAT2.2. This test rig is used for the real-size blade bearings listed in Table 1. The two test bearings (1 and 2) are connected to an intermediate ring (4). A motor turns both outer rings at the same time. It is fixed at the upper bearing's inner ring and acts on the outer ring, A hydraulic system (3) applies a static load to the inner rings. The maximum load capacity is 10 MN. For this test, a contact pressure of 2.5 GPa with a contact width $2b_{51818} = 2.7\ mm$ has been chosen. This equals the contact pressure during the test on the BEAT0.2.





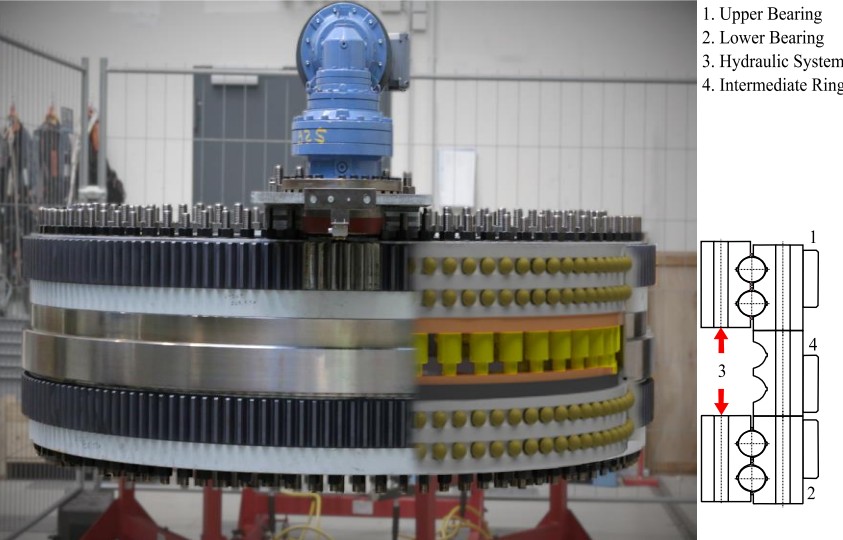

Figure 14: BEAT 2.2 with real-size blade bearings 51818 (Behnke and Schleich (2022))

The bearing manufacturer mechanically cleaned and lubricated the bearings with grease no.1. In the test rig, the bearings are turned for a few times to ensure an equal grease distribution prior to the test program.

### 4.3.4 Test setup for blade bearings type 12480

Figure 15 shows the BEAT6.1 test rig. It makes use of a similar hexapod construction as the smaller BEAT1.1 to be able to apply realistic loads to the tested bearings in six degrees of freedom. However, it has a much higher load capacity with bending moments of up to 50 MNm and room to test bearings with up to 6.5 m in diameter. The rig tests two bearings at the same time in a back-to-back configuration and uses special adapter parts for emulating stiffness properties of hub and blade connection of the bearings. A detailed description of the test rig and its adapter parts can be found in (Stammler, 2020). The tests of the type 12480 bearing can be considered as the most realistic ones in the context of this paper. Due to the usage of a blade adapter constructed with fibre reinforced plastics and t-joint bolted connected at the blade facing flange of the bearings, a rather malleable deformation characteristic is represented. In addition, no scaling is applied to the test program as the simulation results of the IWT-7.5-MW reference wind turbine model are directly representing the loads and operating conditions of this bearing. As the costs and effort related to a real scale blade bearing test on the BEAT6.1 are significant, the bearing was exposed to several test conditions, prior to applying the test program presented in this work. The bearing was exposed to various static and dynamic test conditions to examine the friction torque behavior as well as for model validation purposes. In addition, the bearing was subjected to an endurance test program, which consist of dynamic loads, dynamic and wear critical pitch movements as well as ongoing relubrication according to manufacturer recommendations. The overall test duration is several months. The endurance test program is based on the work of Stammler (2020) and will be presented in another paper in detail. To minimize the risk of interactions of the results of the different test programs, the loads of the test program of this



340    work have been rotated in the corresponding coordinate system by 180°. Hence, traction and compression side of the loaded

bearing changed and both test programs applied the loads to different raceways of the test bearings.

All test rigs mentioned in this paper are located within a test hall with room temperature.

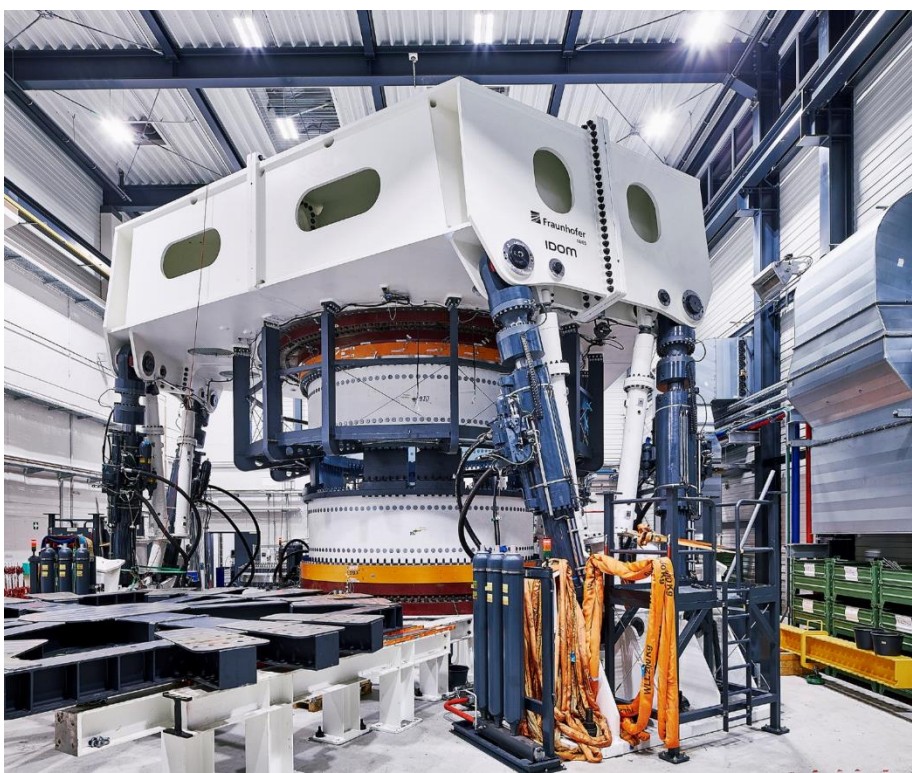

345    **Figure 15: BEAT6.1 test rig (©Fraunhofer IWES/ Marcus Heine)**

### 4.4 Scaling of the test program

The absolute amplitudes from the simulation of the reference turbine must be scaled to the smaller bearings used for this
investigation. The scaling is based on a constant ratio of translatory movement of the contact area $x$ and its load dependent
width $2b$. This x/2b ratio can characterize the oscillating movement of bearings independently of the bearing size. A detailed
350    explanation of the scaling procedure is given in (e.g. Wandel et al. (2022), Stammler et al. (2020), Schwack et al. (2020)). The
scaling calculates according to Eq. (3):

$$\frac{x_{IWT}}{2b_{IWT}} = \frac{x_{scaled}}{2b_{scaled}} \tag{3}$$

The translational travel distance x related to the inner ring of the bearing is calculated using the pitch angle $\theta$ and $\gamma$ and $d_i$
according to Table 1.





$$x = \frac{\theta \cdot (1+\gamma) \cdot d_i}{4} \tag{4}$$

As the load distribution in a blade bearing is heavily influenced by the stiffness of the surrounding structures and the complex external load conditions with five degrees of freedom, it should be assessed using methods like finite element analysis (FEA). For the sake of simplicity, this work uses an approximate calculation of the maximum ball load suggested by Harris et al. (2009). The width of this Hertzian contact $2b$ can be calculated according to Houpert (2001).

Given the different bearing properties, Eq. (3) and (4) can be converted to be able to calculate the pitch angle of the scaled bearings while maintaining the same x/2b ratio as in the simulation. Due to the static load application of the BEAT0.2 the width of the Hertzian contact $2b_{7220}$ of the type 7220 bearings cannot be adjusted during the test and the dynamic and load dependent value of the simulation $2b_{IWT}$ must be converted to a constant value as well. Therefore, the median value of the specific, simulated 10-minute interval $\widetilde{2b}_{IWT}$ is used to calculate the scaled pitch angle $\beta_{7220}$ for a width of $2b_{7220}$ at a constant contact pressure of 2.5 GPa.

$$\theta_{7220} = \frac{di_{IWT}}{di_{7220}} \cdot \frac{(1+\gamma_{IWT})}{(1+\gamma_{7220})} \cdot \frac{2b_{7220}}{\widetilde{2b}_{IWT}} \cdot \theta_{IWT} \tag{5}$$

As the BEAT2.2 test rig has the same limitations with respect to the load application as the BEAT0.2, the same simplifications as in Eq. (5) are also used for the larger blade bearings. Equation (6) how the scaling is performed for these larger bearings.

$$\theta_{51818} = \frac{di_{IWT}}{di_{51818}} \cdot \frac{(1+\gamma_{51818})}{(1+\gamma_{IWT})} \cdot \frac{2b_{51818}}{\widetilde{2b}_{IWT}} \cdot \theta_{IWT} \tag{6}$$

The scaling for the downscaled blade bearings is different to the before mentioned bearing types. The reason for this is the ability of the BEAT1.1 to load the bearings dynamically in five degrees of freedom due to its hydraulic hexapod design. Hence, the bearing specific maximum contact width $2b_{13229}$ is not set to a static value during this test. Instead, the simulated maximum contact pressure resulting in $2b_{IWT}$ at every time step during the simulations is used to calculate scaled load conditions for the smaller bearings. Therefore, the loads during the test at the BEAT1.1 have the same frequency and characteristic as in the simulation but with a smaller magnitude to result in the same maximum contact pressure as in the simulation of the IWT-7.5-164. Equation (7) shows how the scaling is performed for the downscaled blade bearings.

$$\theta_{13229} = \frac{di_{IWT}}{di_{13229}} \cdot \frac{(1+\gamma_{13229})}{(1+\gamma_{IWT})} \cdot \frac{2b_{13229}}{2b_{IWT}} \cdot \theta_{IWT} \tag{7}$$

Figure 16 shows the results of the scaling with respect to the pitch angles for the downscaled blade bearings at the BEAT1.1 for an interval of 600 s of the test program. The graph to the left indicates that the pitch angle of the smaller test bearing becomes slightly larger than the one from the IWT-bearing of type 12480. The plot on the right gives the resulting x/2b ratio from the simulation and from the test program during the same 600 s interval. It is noticeable that the x/2b ratio is not changed by the scaling and stays the same for both bearings.





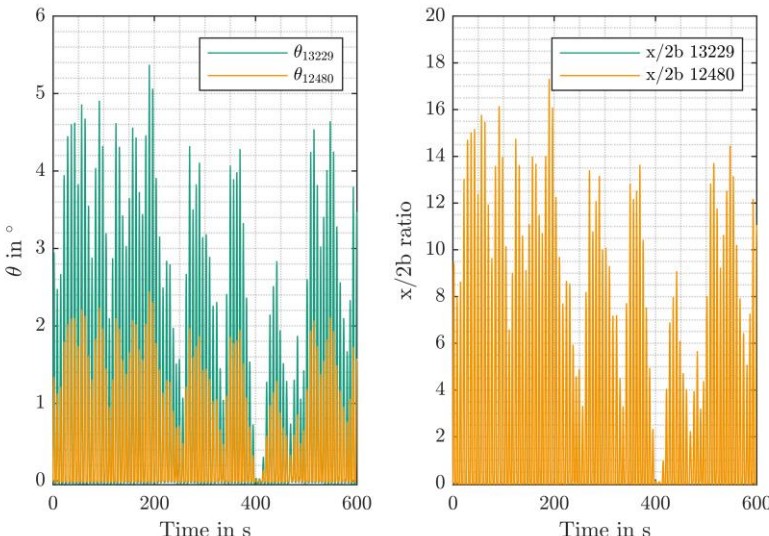

**Figure 16: Scaling of time series data between different bearing sizes; Scaled pitch angles (left) and corresponding x/2b ratio (right)**
**for an interval of 600 s from the test program**

## 5 Test results

This chapter describes the experimental results of this work. After the tests, the bearings are dismantled and analyzed. The characterization of wear damages is based on visual inspection.

### 5.1 Results of type 7220 angular contact ball bearing and type 51818 blade bearing

As the test set up of the small-scale type 7220 bearings and the real scale blade bearings is very similar, this section covers the results of both tests.

After the test on the BEAT0.2, the bearings of type 7220 have shown several signs of wear (see Figure 17). The wear damage is characterized by adhesive damages and fretting corrosion as well as some early abrasive wear regions indicated by a more polished appearance. However, the damage has not been present over the entire contact track. The segments marked by the

black frames in Figure 17 are indicating the angular distance between two neighbored rolling elements. The maximum pitch angles within the test program are close to $\theta_{crit}$ but not larger. Hence, no overlapping of neighbored rolling elements has occurred on the raceway throughout the test and only a small fraction of the raceway has been affected by wear. The distinct lower turning point at about 0° caused by the controller characteristic (see Figure 9) is visible as a slightly shiny line to the left of the wear marks.



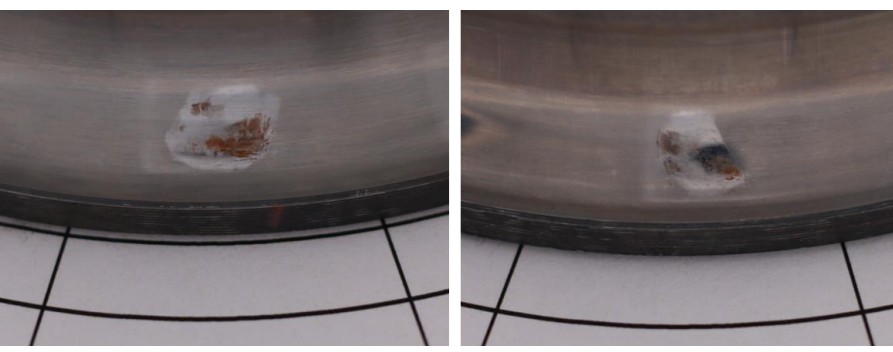


**Figure 17: Wear marks on the type 7220 bearings**

The test on the BEAT2.2 have led to wear damage on the type 51818 blade bearings which has been slightly more pronounced in comparison to the results of the type 7220 bearings. Figure 18 and Figure 19 show examples. Comparing both figures, it becomes evident that the severity of the marks differs. The two marks in Figure 18 are larger. The lower turning point of the times series can be identified on the left side. A corrosion product - probably hematite, due to its brown color - is visible over the entire height of the wear mark. The right end is not clearly defined. The cycles with different amplitudes, which are typical for the test program, have led to this "washed out" result. The marks shown in Figure 19 have a less distinct shape. It is not possible to identify begin, end, or height of the contact track. Areas with pronounced and with less pronounced damage are spread over the circumference of the bearing.

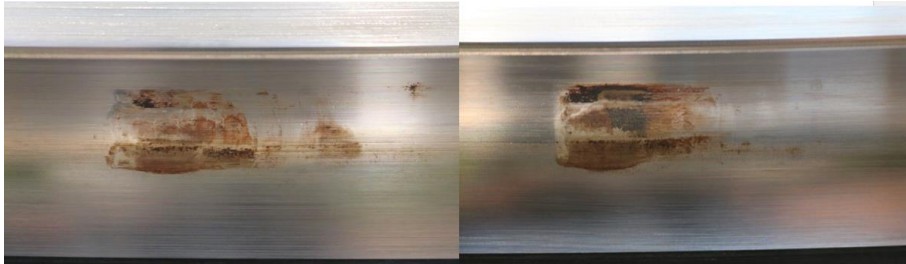


**Figure 18: Severe wear marks on the raceway of 4pBB 51818**

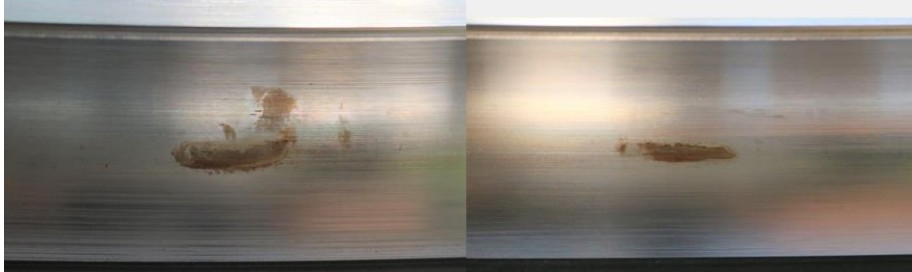

**Figure 19: Moderate wear marks on the raceway of 4pBB 51818**




**5.2 Results of type 13229 and type 12480 blade bearing**

The results of the scaled blade bearings on the BEAT1.1 test rig and of the real scale blade bearings on the BEAT6.1 are described in the following. Compared with the previously described results, especially the results of the scaled blade bearings are quite different. While the test with the grease with the lower base oil viscosity (grease no.1) has resulted in almost no signs of wear, the test with the grease with higher base oil viscosity (grease no.2) has caused comparable wear damages.

Instead of a corrosive damage as an early stage of a false brinelling damage, only a minor change in the surface roughness has

been detectable with grease no. 1 (see Figure 20). However, the regions of the raceway which are subject to movements of the rolling elements are visible and share almost the same appearance as the wear marks of the type 7220 bearings (see Figure 17). The changes in surface properties show the same distinct lower turning point at about 0° caused by the controller characteristics and indicated by the hard stop to the left of the slightly more dull appearing areas. The right end of these areas appears more washed out which again is a consequence of the variable amplitudes of the test program. In addition to the formation of early

wear marks, sections with cross marks can be observed on the scaled blade bearings. The origin of those cross marks is not clarified yet, but it is unlikely that these are results of the tests performed on the bearings.

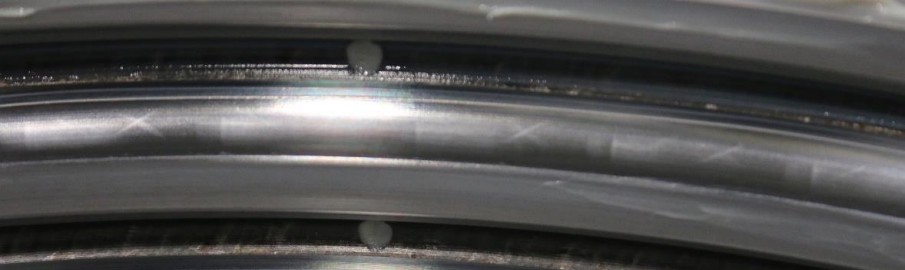

**Figure 20: Wear marks on the raceway of the downscaled blade bearings 13229 lubricated with grease no.1**

In contrast the results of the test with grease no.2 have shown wear marks which are comparable with the ones which have

been obtained from the test with type 7220 and real scale blade bearings (see Figure 21).

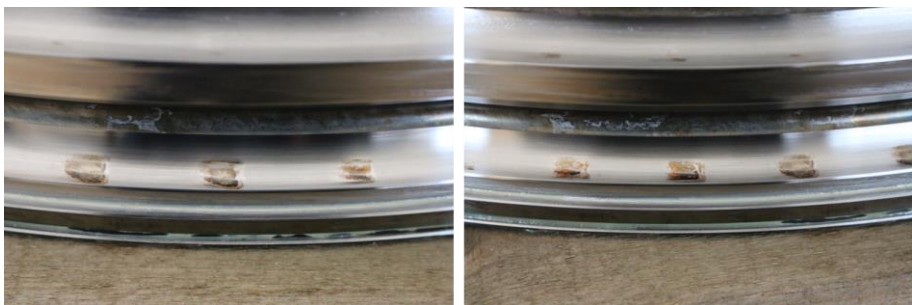

**Figure 21: Wear marks on the raceways of the downscaled blade bearings 13229 lubricated with grease no.2**

The appearance of the wear marks is comparable to the more severe wear marks found on the type 51818 blade bearing. They

share the same overall appearance with the distinct turning point on the right edge and the same corrosive damage. However,





due to the bending moments applied with the BEAT1.1 and the corresponding load distribution in the bearings, the damage is not spread over the complete circumference. Instead, only the highly loaded areas at 0° and 180° have been affected by the wear damage.

For a more complete picture, the test program with dynamic loads and without any scaling was applied to blade bearings with almost 5 m diameter using the BEAT6.1 test rig. The tests of these type 12480 blade bearings revealed the following results: Figure 22 shows some wear marks on the outer ring of the bearing after the test. Several wear marks can be found from the highly loaded traction side areas at about 0° to the less loaded areas at about 90° and 270°. All wear marks have shown early signs of corrosive damage and stages of adhesive wear. However, again the damage does not spread over the entire height of

the contact zone and not over the complete width of the oscillating movement of the test program. In comparison with the other test results, the wear marks are less pronounced as the ones in the type 51818 blade bearings and less severe than in the type 13229 scaled blade bearings equipped with grease no. 2.

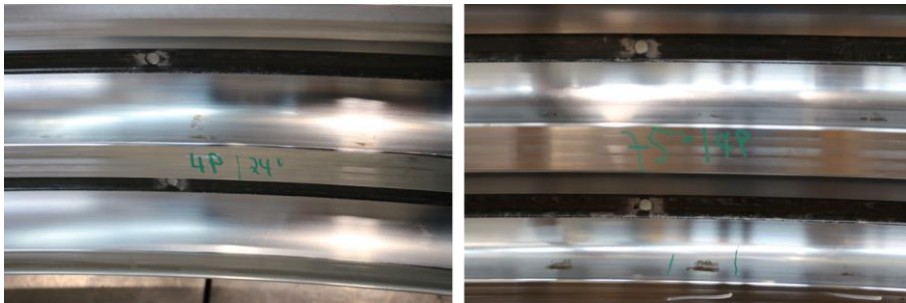

**Figure 22: Wear marks on the raceways of the type 12480 blade bearing outer ring**

**6 Discussion**

While the test results of type 7220 bearings and real size blade bearings of type 51818 show almost similar results, the results from the scaled blade bearings of type 13229 and the real size blade bearings of type 12480 are different. The possible reasons for this difference will be discussed in the following.

The BEAT0.2 and the BEAT2.2 load the bearings with a static axial load. This translates to a constant contact pressure and a

constant contact angle between rolling element and raceway and is comparable to many other experiments conducted and published in the past (e.g. Stammler et al. (2019), Wandel et al. (2023), Behnke and Schleich (2022), Bayer et al. (2023)). On both test benches, wear damages are produced by the execution of the scaled test program. This shows that the scaling mechanism used for the test program is suitable for creating similar and challenging operation conditions for both bearing types and sizes. The major difference of the test program of the scaled blade bearings and the type 12480 blade bearings is that

the loads are not static. Instead, the loads follow the same frequency and general composition as in the simulated reference wind turbine model. Hence, the contact pressure and contact angles are dynamically changing throughout the test execution. This could influence the lubrication conditions in the contact area mainly due to a larger contact area on the raceway caused



by the contact angle variations. In addition, the varying contact pressure could also lead to different rheological behavior of the lubricant used throughout the tests. However, while the test of the scaled blade bearing and grease no. 1 did not lead to
considerable damage, more severe damage was found on the real scale blade bearings of type 12480 after the test at the BEAT6.1 test rig. This indicates that either the scaling between those two test programs changed the effect on damage initiation or design differences between both bearings lead to different relubrication conditions of the contacts. Such differences could be triggered by the cage designs which can affect relubrication.

Another important aspect in the assessment of the test results in this paper is the lubricant. All tests are conducted using the
same grease (grease no.1), which was also used and rated against other commercially available greases for blade bearings in other investigations. In comparison with other lubricants grease no.1 has a good aptitude for preventing wear in critical operation conditions (Schwack et al., 2020). Hence, the grease was also used in this investigation to assess the severity of the presented test program. While the grease was not able to protect the type 7220 bearings and the real scale blade bearings of type 51818 under purely static load conditions, it showed a sufficient performance with the downscaled blade bearings and
dynamic loads. However, in the real scale blade bearings of type 12480, the grease was not able to prevent wear from forming on the raceways. Grease no. 2, results in more severe signs of wear damage after running the identical test profile on the scaled blade bearings. Even though the test program with dynamic loads and constantly changing contact angles and contact pressure probably leads to less harmful operating conditions with respect to wear, not every commercial grease is able to prevent it completely. Grease no. 2 has a significantly higher base oil viscosity than grease no. 1. Its reduced mobility offers less
protection against wear especially at small oscillation angles (Wandel et al., 2022).

Despite the differences of the test results, they show that a rather short time frame of wear critical operation conditions can lead to early wear damages on the raceways of blade bearings. Hence, the composition of 13.7 h wear critical operation without pitch cycles large enough to redistribute the lubricant and protect the bearing has proven to be sufficient to cause wear although it covers only a very small portion of the service life of a wind turbine. This indicates that lubrication runs executed on a
regular basis of for example every 24 h are associated with a risk of not preventing wear. On-demand executions of lubrication runs could not only lower the risk of wear but also be less intrusive with respect to loads and energy output of the wind turbine. However, there is no method or monitoring solution which can trigger on-demand execution of lubrication runs available to the knowledge of the authors. Future work will focus on this subject.

It should be noted that the results of the test program depend strongly on the controller and turbine configuration and could be
very different with another turbine configuration. Therefore, the presented methodology could be used to assess different controller designs with respect to their risk of causing wear in blade bearings. The same applies to the possibility of testing different greases with the proposed test program to assess their suitability for protecting blade bearings from wear.

The results proof that the simple wear rating factor introduced with Eq. (2) as well as the criteria of the selection of the critical sequence of wind conditions can be used to define a wear dedicated test program. In addition, this proofs that wear can be
caused in a rather short amount of time compared with the designed turbine service life.





However, the site-specific wind conditions used in this investigation only cover about one year of measurements. Hence, measurements at another site or over a longer period could lead to even more challenging operation conditions with respect to wear. Future work will address these topics with the goal of developing a more general assessment methodology regarding wear critical operation conditions.

The results also show that test conditions which simplify the operation conditions of blade bearings in terms of static axial load instead of dynamic loads in five degrees of freedom can be more critical. The result of such test could be interpreted in way that the lubricant offers insufficient protection for the application, or the controller and turbine specific operation conditions are causing wear. However, this may not be the case as and a validation for the real application based on these results is not recommended as not all necessary kinematic and rheological aspects can be covered with those simplified test

conditions. Besides this, the simplified tests allow for a relatively simple and fast assessment of different lubricants and the general influence of wind turbine typical pitch movements with variable oscillation amplitudes. When it comes to functional testing of a bearing and controller design, more sophisticated test environments with multiple degrees of freedom and the ability to run dynamic loads should be used.

All wear damages produced in the tests and presented in this paper can be considered as mild wear, which will not directly

affect the functionality of a blade bearing. The question about the influence of such wear damages on the long-term behavior including the initiation of other failure modes and the effect on the lifetime of a blade bearing is subject of ongoing research activities.

## 7 Conclusions

The hypothesis that wear in blade bearings can occur quickly under certain operating conditions was proven through testing of blade bearings, using a program based on site-specific and measured wind conditions, on both scaled and actual size bearing types. The test program consists of several hours of pitch activity based on simulated, stochastic wind conditions, and a specific pitch controller and turbine design. It can be adapted to other turbine and controller designs as well as to other site-specific wind conditions. In addition, the applied scaling methods are verified by applying them to different bearing sizes. Despite the

differences in size and bearing design, the test program leads to comparable wear damages on the raceways. However, the results of the tests with dynamic load conditions suggest that not all aspects of blade bearing specific load conditions relevant for the creation but also prevention of wear damage can be covered with purely static load conditions and simplified test rig designs. Such tests with static loads can nevertheless be of value for rating and comparing operation conditions and lubricants. However, a validation for the application of lubricants and operation conditions should be performed with more sophisticated

test methods and environments. As the test program with its rather short time frame of only 13.7 h has been able to produce wear damage on the tested bearings, the design of pitch controllers and the design of lubrication strategies should not only



depend on short-term intervals such as load time series data. In addition, the medium-term operation conditions of the turbine over hours have to be considered in order to assess the risk of wear related damage of the blade bearings.

**Author contributions**

A. Bartschat: Conceptualization, Methodology, Investigation, Validation, Writing - Original Draft, Project administration, Funding acquisition; K. Behnke: Writing - Original Draft, Investigation; M. Stammler: Writing - Review & Editing, Funding acquisition

**Competing interests**

The authors declare that they have no conflict of interest.

**Financial support**

This research has been supported by the German Federal Ministry for Economic Affairs and Climate Action with the projects "iBAC – Intelligent Bearing Amplitude Control" (grant no. 0324344A), "HBDV - Auslegung hochbelasteter Drehverbindungen" (grant no. 0324303D) and "HAPT- Highly Accelerated Pitch Bearing Test" (grant no. 0325918). The project funding is kindly acknowledged.

**Acknowledgement**

The authors would like to thank the project partners for supporting the projects with knowledge and data. The wind speed measurement data as well as support for developing the controller used for this work were kindly provided by Enercon in the projects HAPT and iBAC. The authors would also like to thank Heinrich Drath and Nils Thormählen for their support in assembling and operating the test rigs for the experiments.

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
