# Peer review of "The effect of site-specific wind conditions and individual pitch control on wear of blade bearings"

_Wind Energy Science, 2023_

## Author Comment (AC1)

Dear reviewer,

Thank you for the comprehensive and helpful review of the proposed paper. Please find my answers and comments below:

**General Comment:**

- As there are three different heights of measurement, it is not clear from which height wind shear and turbulence intensity are considered for simulation. Please add it to the related section

Thanks for pointing this out. I included the following sentence in section 2.1 line 99:

*As the IWT7.5-164 reference turbine has a hub height of 119.3 m all measurement data related investigations in the following are focusing on the measurement height of 119 m.*

- Please explain why six seeds are used for each wind speed or reference it. According to IEC 61400-1, in the ultimate analysis, six seeds were used at a (below-rated wind speed – 2 m/s).

The reason for simulating six seeds per 10-minute interval is the same as in the IEC standards. According to IEC 61400-1 section 7.5 at least six 10-minute stochastic realizations of turbulent wind input are required for each mean, hub-height wind speed to ensure statistical reliability. Hence, the reference will be updated in line 174 of the paper.

**Specific comment:**

1. In Section 2.1, please add the goodness-of-fit of the Weibull distribution.

The fit of the Weibull-distribution shall only give some reference information on the characteristics of the site. Hence, the shape parameters of the fit are given to be able to do a more detailed comparison with the shape parameters suggested for the different site classes according to IEC 61400-1. As all investigations and simulations mentioned in the paper are referring to the raw measurement data the goodness of the fit has no impact on any of the results. However, for the sake of completeness the RMSD between data and fitted function was calculated and the legend entry of figure 1 was updated accordingly. In addition, the statement in line 101 was adjusted to be less confusing.

*The site is generally characterized by rather low wind speeds.*

2. The phrase "the wind speeds do not correlate well with any of the IEC wind classes" in Section 2.1 needs to be quantified.

You are right. When referring to a correlation a measure for this would be more comprehensive. However, as the focus is not set on the differences between measurements and distribution properties given in the standards but on how site-specific wind conditions and especially the medium-term intervals can affect wear in blade bearings, no quantitative measures are suggested. For the sake of completeness, the shape parameters for the fitted distribution are given in the figure for more in depth investigations.

3. It is not clear what the height of measurement is in figures 1 and 2. Please add the height in both figures.

You are right. The figure description was adjusted accordingly.

4. In figure 2, the last bin does not have a mean plus (red asterisk) value.

Thanks for pointing this out. Due to the lack of data points in the last bin, no standard deviation values are calculated. Hence, the last bin was deleted from figure 2 and figure 3.

5. In Section 2.1 and related to Figure 3, it seems that heights of 119 and 139 are used for the calculation of the shear exponent, and the projected wind speed results on 159 meters are compared with measurements. If that is the case, please include the results of the comparison. Otherwise, explain more about the process and comparison with measurement.

The exponent α which leads to the lowest error between power law function and measured data is used as an estimation for the shear. Figure 3 shows the obtained and bin sorted results for each dataset. The measurements at a height of $z_{hub} = 119\ m$ are used as $V_{hub}$ in the context of the power law function (Eq. 1). All three measurements at 119 m, 139 m and 159 m are used to define α. I updated the sentences starting from line 113 accordingly:

*Therefore, the wind speed measurements for each time step and for the three different heights are used to fit the power law function with a hub height of z_hub=119 m in Eq. 1*

*The exponent α which leads to the lowest error between power law function and measured data at all three heights is used as an estimation for the shear.*

6. In the phrase "While the shear induces more pitch activity, the low TI leads to less variation in the oscillations of the bearings. It is therefore reasonable to assume a more wear-critical operation under the measured wind speeds in comparison to IEC classes." It is not clear how this judgment was reached. Please explain more.

The foregoing sentences beginning in line 123 are explaining, why the authors came up with this assumption. However, to give more context with respect to the risk of wear I included the following extension to this section of the text:

*Without much variation in the oscillations and without larger movements due to lubrication runs the same spots of the raceways are in repeated contact with the rolling elements and the possibility of starved contact lubrication conditions is rising.*

7. Wrong referred In line 238, Eq. (1) should be changed to Eq. (2).

Thanks for noticing. The reference was changed to the correct equation.

8. It seems Equation 5 has the wrong term. It has to be rewritten.

Again, thanks for noticing this detail. Equation 5 is correct, but equation 6 and 7 have incorrect terms. I corrected the equations as follows:

$$\theta_{51818} = \frac{di_{IWT}}{di_{51818}} \cdot \frac{(1 + \gamma_{IWT})}{(1 + \gamma_{51818})} \cdot \frac{2b_{51818}}{\widetilde{2b}_{IWT}} \cdot \theta_{IWT}$$

$$\theta_{13229} = \frac{di_{IWT}}{di_{13229}} \cdot \frac{(1 + \gamma_{IWT})}{(1 + \gamma_{13229})} \cdot \frac{2b_{13229}}{2b_{IWT}} \cdot \theta_{IWT}$$

---

## Author Comment (AC2)

Dear Jonathan Keller,

Thank you for the comprehensive and helpful review of the proposed paper. From my point of view the quality of the paper has been improved by incorporating the suggested changes. Please find my answers, comments, and changes below:

**General Comments:**

I tried to understand your general comment to the proposed paper and felt that some parts of it can indeed imply a misleading understanding of what the work is about. In fact, the paper is not intending to present control strategies which are lowering the risk of wear or to do a comprehensive comparison between different controllers and different site-conditions. Instead, the paper can be better understood as a case study aiming at assessing and testing the wear risk of an individual pitch-controlled turbine under the influence of site-specific wind conditions. Therefore, some methods are presented which can be used to evaluate the wear risk by testing and which revealed that wear can occur in a rather short time frame of only several hours of operation. To avoid misleading expectations about the content we incorporated several changes as suggested in the specific comments section. In addition, the abstract has been changed from line 12 onwards to a less universal or comparative assessment of the wind and operating conditions:

*This work analyses exemplary wind and operating conditions of one specific site regarding their influence on wear in blade bearings. It is based on measured wind conditions and the modelled behavior of the individual pitch-controlled IWT-7.5-164 reference wind turbine with respect to its pitch activity.*

I will try to answer some of your more general comments in the following.

- After giving some thought about trying to sum up the article in a single sentence, I personally would conclude that the work shows that "Pitch bearing wear can quickly result from individual pitch control". Can that be as broadly implied as written? Or does it entirely depend on the nature of the site, turbine, and such?

Your takeaway is correct. However, as discussed in section 6 it should be noted, that while it is possible to derive this more generalized statement, different site-specific wind conditions and controller characteristics can lead to a different result. Hence, the proposed methodologies can be used to assess the individual wear risk for example by running scaled tests.

- Can we quantify in anyway what we thought the risk of the conditions might have been, compared to what they were demonstrated to be?

At this stage, there is no documented and straight forward way of assessing and quantifying the wear risk during the design process of the pitch controller. In addition, and as mentioned in the discussion section I am not aware of an on-demand lubrication run strategy at this point. Future work will address this and will provide suggestions for this.

- Then again, maybe a more important point is that the actual site-specific characteristics might give different results than the standard IEC classes – maybe the latter would yield little to no wear?

Despite the different characteristics of this exemplary site with respect to turbulence intensity and vertical shear in comparison with the IEC standards, especially the lack of a suggestion on medium-term behavior of the wind makes it difficult to assess the wear risk with respect to the IEC classes. The order of occurrence of operating conditions over the periods of hours and days is the challenge of determining the wear risk.

**Specific Comments:**

- Title: I have to wonder, is the aspect of "site-specific" worthy of being in the title? It is certainly true that this examination was accomplished using the particular wind characteristics of an actual site and for a reference (or representative of the state-of-the-art) large, modern wind turbine. But, I believe those to be secondary. Would the authors agree? It was also not shown/quantified (or at least I didn't pick up on it) that the site-specific conditions were any more problematic in terms of wear than standard IEC wind classes and NTM. I think a simpler title like "The effect of individual pitch control on blade bearing wear" is more accurate and better in line with the Test Results, Discussion and Conclusions.

The site-specific wind conditions are characterized by a specific composition of wind speed, turbulence intensity, vertical shear. The important aspect of those wind conditions with respect to wear damage of blade bearings is the order of occurrence and overall duration of wear critical periods. This aspect is usually not covered by design processes of pitch controllers or operating strategies of blade bearings, but as a conclusion, should be looked at. Hence, I believe that the term "site-specific" is of importance for this work. However, to lower the risk of misleading impressions on the context of the paper the abstract has been changed as mentioned in the general comment section.

- Abstract: I think it is common that the Abstract should be a single paragraph. I also think what is now the second paragraph can be improved to be more specific as in "This work demonstrates that a site-specific wind and pitch controller conditions can quickly result in pitch bearing wear." That is, the article doesn't generally analyzes (plural) "wind and operating conditions", just one particular combination. OK, across wind speeds yes, but really seems to be specific to this particular site. I also recommend adding the adjective "mild" to wear in the Abstract to be in line with the last sentence in Section 6.

Thanks for the suggestions and you are right with some misleading formulations. In addition to the changes mentioned in the general comment section I implemented the term "mild wear".

- 1 Introduction: There are some minor grammatical changes needed here. In line 29, I recommend "…for evaluating the wear risk to pitch bearings…" This paragraph is also only 2 sentences long. I believe it can be combined with the next paragraph beginning on line 31.

The short paragraph beginning in line 29 is now combined with the former one.

- 1 Introduction: The sentence describing theta_crit in lines 32-33 can be improved, specifically "gets in touch". I believe per DG03 and per Rumbarger and Jones that theta_crit is "the relative angle of rotation of the raceways for which the portion of the raceway stressed by one rolling element touches—but does not overlap—the raceway that is stressed by adjacent rolling elements."

The description of $\theta_{crit}$ has been updated to match the formula expression of the DG03 as follows:

$\theta_{crit}$ *is the oscillation angle of the inner raceway relative to the outer raceway for which the center of the contact of one rolling element moves to the initial center of contact of the adjacent rolling element.*

- 1 Introduction: The sentence on lines 36-37 also says "Tests on small scale bearings performed by showed…", which reads as though a name and/or reference is missing, or "performed by" can be deleted. I suggest the former though.

Thanks for noticing. "performed by" has been deleted.

- 1 Introduction: I will admit I got a bit confused reading lines 40 – 47 and "the work related to this paper focuses", lines 53-54 and "aim of this test program is to", and lines 57-58 and "The study in this paper is not". I kept thinking that they were describing this manuscript, when I think they are describing nearby citations. I recommend these be past tense and/or "this" changed to be more specific. The statement about this manuscript does not come until lines 76-79. Generally speaking, I found lines 45-75 a bit hard to follow.  It might also help to begin a new paragraph with "Stammler (2020)…" on line 47 and maybe trim some of this material in lines 47-75. At a minimum, I would suggest defining the acronyms for HBDV, HAPT, and iBAC, and if possible adding a citation. I'm not sure simply mentioning them is worthwhile if no other information is given.

To improve the argumentation and the readability of the introduction of the paper based on your comments, the text has been changed and partly rearranged from line 35 to 79. In addition, I added a reference to the financial support section in line 64 which holds all necessary information on the mentioned research projects.

[revised manuscript text omitted]

- 2.1 Wind data characterization: The colors in Figure 1 (bottom) were a bit hard to sort out. I might suggest that because the Weibull distribution fit is not mentioned in the text, it can be removed from the figure – or do the listed a and b values quantify that the measured wind speeds "do not correlate well with any of the IEC wind classes"?

I changed the colors and appearance of the bars to be translucent to improve the visual impression. The fit of the Weibull distribution in general shall help to see the differences of the standard IEC classes. The shape parameters are given for the sake of completeness and can be used for a more detailed comparison of site characteristics and IEC wind classes. However, this level of detail is not needed for the argumentation within this paper.

- 2.2 Wind data and turbine operation conditions: I had difficulty in following the descriptions in this section. At first, I thought this section was intended to describe the process by which wear-critical operations are determined, but then the first sentence in Section 3 says that is the point of Section 3. I think Section 2.2 maybe just gives an example of a particular interesting/likely to induce wear condition, but it's probably longer and more convoluted than needed at several pages and 3 figures.

Section 2.2 describes the fundamental assumptions and the methodology of looking at the measured wind data to define the critical sequence of wind conditions. Hence, you first thought is correct. As this

an essential step, I think the level of detail is appropriate. Section 3 uses the output of the described analysis as an input for running simulations which aim at closing the gap between wind measurements and turbine reaction like loads and pitch movement.

- 4.1 General description of the test program and simulation results: I think some of my confusion about Section 2.2 is not surprising, as the first sentence in Section 4.1 says "To evaluate if the critical sequence described in Chapter 2 is in fact critical in terms of wear..." That is, what is the difference between "critical sequence" and "critical (sequence) in terms of wear"? I found the process description and Figures 10 and 11 helpful here – it just felt like it took a long time to get to this point.

The wording in lines 199 – 200 has been changed for more precise description and to be more in line with the process described in Figure 10:

*A test program has been developed based on the critical sequence of wind conditions and using the simulation results of Chapter 3.*

- 6 Discussion and 7 Conclusions: The statement in 6 "While the test results of type 7220 bearings and real size blade bearings of type 51818 show almost similar results, the results from the scaled blade bearings of type 13229 and the real size blade bearings of type 12480 are different" seems to conflict with the statement in 7 that "Despite the differences in size and bearing design, the test program leads to comparable wear damages on the raceways."

Thanks for pointing this out. The statement in the conclusions section has been slightly changed by deleting the adjective comparable.

*Despite the differences in size and bearing design, the test program leads to wear damages on the raceways. However, …*

- 6 Discussion and 7 Conclusions: In 7 Conclusions and to my earlier questions about the importance (or not) of "site-specific" conditions to pitch bearing wear, where it is stated in the first sentence on line 515 that "…wear…can occur quickly under certain operating conditions…" I do agree that this has been shown but more than anything what I do not have a clear picture of is exactly what "certain" means here. What exactly are the characteristics that lead to wear? Can we say, or is every site a snowflake? I take it the IPC characteristics and maybe the turbulence and/or shear are the most important things? Or maybe the main point is that the article proposes a process by which a particular set of characteristics can be assessed for a given turbine/controller/bearing, but then again leads me back to the thinking that every site is a snowflake. That is, the characteristics of any given site must be known and then tested to see the extent to which wear may result and then the IPC potentially changed, which is implied where it says "It can be adapted to other turbine and controller designs as well as to other site-specific wind conditions."

I added a sentence describing the "certain" operating conditions for more comprehensive read of the conclusions section:

*The medium-term, critical sequence of wind conditions characterized by very minor changes in the wind speeds, low turbulence intensity and high shear leads to pitch movements without much variation in oscillation amplitude and frequency and thousands of cycles without being interrupted by effective lubrication runs. The hypothesis that wear in blade bearings can occur quickly under these operating conditions was proven through testing of blade bearings, using a program based on site-specific and measured wind conditions, on both scaled and actual size bearing types.*

In general, you are right with the snowflake theory. Another site could potentially have even more critical wind conditions. The opposite is also possible, and the wear risk can be lower. However, the paper shows that some simple assumptions on potentially wear triggering operating conditions hold true. Hence, the findings can be used for developing test programs which can be used to assess or validate the risk of wear for a given system based on controller, turbine, bearing, lubricant, and site characteristics. In addition, the validated assumptions with respect to wear critical operating conditions can be used to develop more general and universal means for rating the wear risk in the design process or to develop on-demand lubrication run strategies.